# Wide-Targeted Semi-Quantitative Analysis of Acidic Glycosphingolipids in Cell Lines and Urine to Develop Potential Screening Biomarkers for Renal Cell Carcinoma

**DOI:** 10.3390/ijms25074098

**Published:** 2024-04-07

**Authors:** Masamitsu Maekawa, Tomonori Sato, Chika Kanno, Izumi Sakamoto, Yoshihide Kawasaki, Akihiro Ito, Nariyasu Mano

**Affiliations:** 1Department of Pharmaceutical Sciences, Tohoku University Hospital, 1-1 Seiryo-machi, Aoba-Ku, Sendai 980-8574, Japan; mano@hosp.tohoku.ac.jp; 2Faculty of Pharmaceutical Sciences, Tohoku University, 1-1 Seiryo-machi, Aoba-Ku, Sendai 980-8574, Japan; 3Department of Urology, Tohoku University Graduate School of Medicine, 1-1 Seiryo-machi, Aoba-Ku, Sendai 980-8574, Japan; tomonori4659@uro.med.tohoku.ac.jp (T.S.); izumi.sakamoto.e3@tohoku.ac.jp (I.S.); kawasaki@uro.med.tohoku.ac.jp (Y.K.); itoaki@uro.med.tohoku.ac.jp (A.I.)

**Keywords:** glycosphingolipids, liquid chromatography/tandem mass spectrometry (LC–MS/MS), renal cell carcinoma, disialosyl globopentaosylceramide (DSGb5), targeted lipidomics

## Abstract

Glycosphingolipids (GSLs), mainly located in the cell membrane, play various roles in cancer cell function. GSLs have potential as renal cell carcinoma (RCC) biomarkers; however, their analysis in body fluids is challenging because of the complexity of numerous glycans and ceramides. Therefore, we applied wide-targeted lipidomics using liquid chromatography–tandem mass spectrometry (LC–MS/MS) with selected reaction monitoring (SRM) based on theoretical mass to perform a comprehensive measurement of GSLs and evaluate their potency as urinary biomarkers. In semi-quantitative lipidomics, 240 SRM transitions were set based on the reported/speculated structures. We verified the feasibility of measuring GSLs in cells and medium and found that disialosyl globopentaosylceramide (DSGb5 (d18:1/16:0)) increased GSL in the ACHN medium. LC–MS/MS analysis of urine samples from clear cell RCC (ccRCC) patients and healthy controls showed a significant increase in the peak intensity of urinary DSGb5 (d18:1/16:0) in the ccRCC group compared with that in the control group. Receiver operating characteristic analysis indicated that urinary DSGb5 could serve as a sensitive and specific marker for RCC screening, with an AUC of 0.89. This study demonstrated the possibility of urinary screening using DSGb5 (d18:1/16:0). In conclusion, urinary DSGb5 (d18:1/16:0) was a potential biomarker for cancer screening, which could contribute to the treatment of RCC patients.

## 1. Introduction

Renal cell carcinoma (RCC) accounts for 2 percent of all cancers worldwide, and its incidence varies widely among countries [1,2,3]. RCC is a common and heterogeneous form of kidney cancer that accounted for nearly 430,000 new cases annually worldwide in 2020 [4]. Metastatic RCC is associated with poor prognosis and limited treatment options [5]. The main treatment options for RCC are radical nephrectomy or surgery to preserve renal function (partial nephrectomy) using molecular-targeted agents and immune checkpoint inhibitors in cases of metastasis [6,7,8]. One of the characteristics of RCC is its high potential for malignancy. Approximately 33% of RCC patients have metastatic disease at the time of initial presentation, and approximately 40% have metastatic disease after radical nephrectomy [9,10]. The lungs account for the highest percentage of metastatic sites (approximately 75%); however, RCC is also known to spread to various other sites, including lymph nodes, soft tissues, bones, and the liver. Therefore, early detection and treatment of RCC are required. The diagnosis of RCC is primarily based on diagnostic imaging, particularly computed tomography [11]. The pathogenesis of RCC, including its mechanism of metastasis, remains unclear. To achieve early detection and treatment of this disease, it is desirable to search for biomarkers useful for screening and predicting recurrence and metastasis and to understand the molecular mechanisms of pathogenesis in detail [12,13,14].

Glycosphingolipids (GSLs) are a group of substances composed of glycans and ceramides (Cers) and are associated with RCC (Figure 1) [15,16]. Glycans are mainly composed of glucose (Glc), galactose (Gal), *N*-acetylglucosamine (GlcNAc), *N*-acetylgalactosamine (GalNAc), *N*-acetylneuraminic acid (NeuAc), and fucose. NeuAc, also known as sialic acid, has an acetylated amino group at position 5 of neuraminic acid and is the most abundant naturally occurring sialic acid, which is a modification of neuraminic acid. Hereafter, the Symbol Nomenclature for Glycans (SNFG) notation (Figure 2) is used to describe glycans [17,18]. Glycans exist in living organisms in the form of glycolipids, such as GSLs, glycoproteins, proteoglycans, and free glycans. Glycans not only exist in the physical structure and membrane organization of cells but also have other functions, such as intercellular adhesion, signal transduction, and antigen recognition. Cer is composed of sphingoid bases and fatty acids (FAs), and several pathways are involved in its biosynthesis, including de novo, sphingomyelin, and salvage/recycling pathways [19,20,21,22]. The major sphingoid base of Cer in mammals, including humans, is 18-carbon sphingosine (d18:1). By contrast, FAs have chain lengths ranging from 14 to 30 carbon atoms and varying numbers of unsaturated bonds, forming a diverse group of molecules [23,24,25]. GSL biosynthesis proceeds with the addition of Glc and Gal to Cer and elongation of the glycan chain, branching into a pathway called the globo series when another molecule of Glc is attached, lacto series when GlcNAc is attached, and ganglio series when NeuAc is attached (Figure 3).

Although GSLs are expressed in various tissues, their structure and function are highly variable in different types of cancer and are known to underlie the invasive and metastatic potential of tumor cells; similarly, several associations with GSLs have been reported in RCC [23]. In the globo series, there are reports mainly on MSGb5 and disialosyl globopentaosylceramide (DSGb5), whose expression levels have been linked to RCC [26]. DSGb5 inhibits the cytotoxicity of natural killer cells via Siglec-7 [27] and promotes the migration of RCC cells, and DSGb5-high-expressing RCC shows more microvascular invasion [28]. These changes in gene expression may also be observed in RCC [29]. In a recent study, GalNAcDSLc4-positive RCC patients were reported to have a higher risk of metastasis at diagnosis and during follow-up [30]. GM3, GM2, and GM1 have been reported in the ganglio series. GSL, especially GM3, interacts with integrin receptors, inhibits signal activation through Src family kinases or G proteins, and blocks cell adhesion to the matrix. Regarding the association with RCC, GM3 is also expressed in normal renal tissues [31]. In tumor tissues from metastatic cases, long-chain GSLs, such as MSGb5, DSGb5, and GalNAcDSLc4, were increased, whereas in tumor tissues from cases with a good prognosis, long-chain GSLs did not increase, and only GM3 was increased, suggesting that GM3 may be expressed inversely to long-chain GSLs [32]. In addition, the mRNA levels of neuraminidase 3, an enzyme that hydrolyzes gangliosides, were higher in RCC tumor tissues than in non-tumor tissues, and overexpression of the gene resulted in decreased GM3 expression and increased lactosylceramide expression [33]. Thus, GM3 expression may be decreased in contrast to the increase in RCC malignancy; in GM2, which is synthesized from GM3, the gene encoding the synthetic enzyme was overexpressed in RCC tumor tissue [34], and motility and invasive capacity were decreased in RCC cells in which the gene was knocked down [35]. It has also been reported that GM2 expression was confirmed in 83% of RCC tumors from patients with clear cell type, the most common histological type of RCC [36].

As described above, there is an association between GSLs and RCC in each pathway of RCC. However, because of the GSL diversity of molecular species and difficulty in identification and quantification owing to the small number of samples, comprehensive GSL analysis in an ideal study has not been performed, and the existence of RCC-specific GSLs remains unknown. However, accurate quantification and comprehensive qualitative analysis of GSLs in RCC cell lines and bodies have not yet been performed. In addition, the diagnostic performance of GSLs as potential biomarkers for RCC has not yet been evaluated. Therefore, this study aimed to comprehensively analyze GSLs to identify biomarkers that can lead to the detection of RCC. In this study, we aimed to fill these gaps using liquid chromatography–tandem mass spectrometry (LC–MS/MS), a powerful analytical technique that can provide both qualitative and quantitative information about the components in a sample [10,37,38,39,40]. LC–MS/MS is an analytical technique that combines the separation capabilities of LC with the detection capabilities of MS/MS to obtain more specific and sensitive data [41,42,43]. In this study, we performed a comprehensive analysis of GSLs in RCC cell cultures and urine of RCC patients using LC–MS/MS in selected reaction monitoring (SRM) mode. SRM is a quantitative method that detects and quantifies specific analytes by selecting their precursor and product ions in a mass spectrometer [44,45]. SRM has several advantages over other methods, including high sensitivity, specificity, accuracy, and reproducibility. In this study, we developed an SRM method for the simultaneous analysis of 240 transitions of GSLs [46], including DSGb5 and other RCC-related GSLs, in RCC cell cultures and urine samples. In the wide-targeted measurement [47,48,49,50,51], the theoretical masses of GSLs, not only those already reported but also those expected to exist, were calculated, and LC–MS/MS analysis was performed based on them. An analytical method for GSLs was developed. Using this analytical method, GSLs extracted from the cultured cells were measured. We optimized chromatographic and mass spectrometric conditions, validated the performance of the method, and applied it to the analysis of RCC cell cultures and urine samples. First, we used the ACHN cell line, a metastasis-related cell line, to identify the GSL species for screening RCC. We compared the GSL profiles of RCC and normal kidney cell cultures. We analyzed urine samples from RCC patients and healthy controls and evaluated the screening performance of GSLs, especially DSGb5, for RCC detection.

## 2. Results and Discussion

### 2.1. Optimization of LC–MS/MS for Wide-Targeted GSL Analysis

To comprehensively investigate the relationship between GSLs and RCC, we assumed the existence of 240 GSLs consisting of 10 glycans and 24 FA chains known from the literature. We used hypothetical SRM conditions, where Q1 is *m*/*z* based on the theoretical mass calculated from the structure, and Q3 is *m*/*z* 290 derived from sialic acid (Figure 4). To develop an LC–MS/MS method for wide-targeted GSL analysis, it is necessary to optimize MS/MS conditions for comprehensive analysis. In this study, a widely targeted SRM analysis, which enables highly selective and comprehensive analysis, was used, and the conditions were investigated. For MS/MS optimization, mass spectra were obtained using the infusion method with GD1a (d18:1/18:0) standard solution. In the negative ion detection mode, a peak corresponding to a deprotonated molecule ion [M-2H]^2−^ was observed at *m*/*z* 918 as a precursor ion, and a product ion derived from NeuAc was observed at *m*/*z* 290 as the product ion (Figure 5). Both SRM and ionization parameters were optimized (Appendix A). These parameters were used in all the subsequent analyses. In particular, the collision energy (CE) value, which is responsible for fragmentation efficiency, was changed depending on the number of NeuAc molecules because it is affected by the charge number [44]. To separate and measure a variety of GSLs in LC, a gradient-based LC system was constructed using a Nexera UHPLC system with an L-column 3 C18 [44], and the gradient LC conditions were determined, as shown in Appendix A.

Next, the linearity of the MS/MS response was confirmed using LC–MS/MS. We found that the linearity of the response of GD1a (d18:1/18:0) in the range from 0.1 to 1 μg/mL was 0.99937 (Appendix A), suggesting that any change in this range can be analyzed semi-quantitatively in a concentration-dependent manner.

The results of redissolving solvents were examined (Appendix A). Ethanol solution (75%) yielded the best results (95.77%), followed by ethanol (37.95%), methanol (35.50%), and low results for isopropanol (6.29%). Because GSLs have a highly hydrophilic sugar chain and a highly hydrophobic Cer moiety, it is thought that a mixture of water and ethanol gives good results.

Deproteinization and lipid extraction were performed using a combination of the modified Bligh and Dyer method [52,53] and protein precipitation (PP) with acetonitrile. Examination of the recovery of GD1a (d18:1/18:0) using this method revealed that more than 99% of it was recovered from the upper layer (Appendix A).

The matrix effects of GD1a (d18:1/18:0) were investigated. After pretreatment with 10 ng/mL GD1a (d18:1/18:0) standard solution, ACHN cell medium, and ACHN cell medium spiked with 10 ng/mL GD1a (d18:1/18:0) standard solution, the cells were analyzed according to the method described above. The matrix factor (MF) was evaluated by calculating the peak areas of the analytes [44,54,55,56]. The MF of GD1a (d18:1/18:0) in the cell culture medium was 106% (Appendix A); therefore, this method was reliable for quantification.

### 2.2. Measurement of GSLs in Cultured Cells

The developed method was applied to cell and medium GSL analysis to search for metastasis-related GSLs in ACHN cell lines by comparing them with HK-2 cells as a control cell line. In subsequent measurements, the area ratio was calculated by dividing the peak area of each GSL obtained from the sample measurement by the peak area to analyze the relative expression of GSLs. HK-2 cells derived from the human proximal tubular epithelium were used as normal cells. Typical chromatograms are shown in Figure 6.

All peaks in the cells and medium are listed in Appendix A, and the numbers of peaks in both the cells and medium are summarized in Appendix A. The GSL composition results obtained using LC/MS/MS were novel compared with previous results [26,27,28,29,30]. In HK-2 cells, 52 components were not detected, whereas only two were not detected in ACHN cells (Appendix A). In the HK-2 and ACHN media, the number of undetected components was 71 and 68, respectively (Appendix A). The mean GSL peaks were calculated as the ratio of the peaks detected in ACHN and HK-2 cells. In cells, most of the peaks (223 components) of ACHN cells increased (Appendix A). However, in the medium, only 82 peaks of HK-2 cells were observed. The intense peaks in the medium were expected to be candidate biomarkers; therefore, the intense peaks were identified. Consequently, four peaks were found to be the peaks with more than 2,000,000 counts (Appendix A). These peaks are listed in Table 1. DSGb5 at *m*/*z* 984.5, which showed the third intense peak, increased by 459% in ACHN vs. HK-2 (Table 1). However, the ratio of the other three GSLs did not increase as that of DSGb5. Thus, it was hypothesized that DSGb5 (d18:1/16:0) would also be a screening biomarker for human RCC.

### 2.3. GSL Urinary DSGb5 (d18:1/16:0) Analysis in RCC Patients and Healthy Controls

Finally, as a pilot study, we analyzed urinary DSGb5 (d18:1/16:0) levels to estimate the performance of urinary screening for RCC. The urine test results are shown in Figure 7. The urinary DSGb5 (d18:1/16:0) peak was weaker than that in the cell and medium analyses (Figure 7a and Table 1). However, the peak of DSGb5 (d18:1/16:0) was detected in all the patients in this study (Appendix A). This is the first report of urinary DSGb5 results that differs from those of previous reports [26,27,28,29,30]. Urine density is affected by various characteristics, and creatinine concentration correction is generally used. The corrected peak intensity of DSGb5 (d18:1/16:0) in the urine of RCC patients was significantly higher than that in the urine of healthy controls (Figure 7a). Finally, screening performance was estimated using ROC analysis. The AUC value was 0.8810. The estimated screening performance was very high (Figure 7b) and almost the same as that in previous reports for some metabolite combinations. However, the average age of the subjects did not match (Appendix A). Therefore, more age-matched samples must be recruited to validate the performance of urinary DSGb5 levels.

## 3. Materials and Methods

### 3.1. Chemicals and Reagents

GD1a (d18:1/18:0) was purchased from Avanti Polar Lipids Corporation (Birmingham, UK). Ammonium formate and methanol were purchased from FUJIFILM Wako Pure Chemical Corporation (Osaka, Japan). Chloroform (Reagent Grade 1), ethanol (Reagent Grade 1), formic acid (HPLC grade), and isopropanol (LC/MS grade) were purchased from Nacalai Tesque Company (Kyoto, Japan). Acetonitrile (HPLC grade) was purchased from Kanto Chemical Co. (Tokyo, Japan). Acetic acid was purchased from Tokyo Kasei Kogyo Co. (Tokyo, Japan). Ultrapure water was purified using Puric-α-01 from Organo Corporation (Tokyo, Japan).

### 3.2. LC–MS/MS Equipment

The LC–MS/MS instrument used was a QTRAP6500 (SCIEX, Framingham, MA, USA) quadrupole linear ion trap hybrid tandem mass spectrometer with an ESI probe attached to the ion source and connected to an ultra-high-performance liquid chromatograph (Nexera, Shimadzu, Kyoto, Japan). Measurements were performed in the negative ion mode. Analyst 1.6.2 (SCIEX) and MultiQuant version 2.1.1 (SCIEX) were used for analysis and peak area integration, respectively. JMP Pro version 17.1 software (SAS Institute Inc., Cary, NC, USA) was used for statistical analysis.

### 3.3. Optimization of MS/MS Conditions for Targeted GSL Analysis

The SRM transitions that produced the most intense precursor and product ions were investigated using 1 μg/mL of GD1a standard solution at a flow rate of 10 μL/min. The CE, cell exit potential, and declustering potential were optimized under SRM conditions. The ion source parameters in MS/MS were optimized using the flow injection analysis of GD1a (d18:1/18:0). The ion source parameters, collision gas (CAD), turbo gas (GS1), nebulizer gas (GS2), and ion spray voltage (ISV) were optimized.

### 3.4. LC Conditions for Targeted GSL Analysis

Mobile phases A and B were 28% aqueous ammonia solution/water (0.1:100, *v*/*v*) and 28% aqueous ammonia solution/acetonitrile/methanol (0.1:66.7:33.3, *v*/*v*/*v*), respectively, in the gradient flow mode. An L-column3 C18 (2.1 mm i.d. × 150 mm, CERI, Tokyo, Japan) was used as the column, and the column temperature was set at 40 °C [44]. The composition of mobile phase B was increased from 40% to 100% from 0 to 40 min and from 40 to 45 min at 100%.

### 3.5. Setting Virtual SRM Conditions

The theoretical masses of 240 GSLs consisting of 10 glycans and 24 FA chains were calculated, and the SRM analytical conditions were set.

### 3.6. Preparation of Standard Solution of GD1a (d18:1/18:0)

For GD1a (d18:1/18:0), 100 µg/mL of standard stock solution was prepared in a mixture of water/ethanol (1:3, *v*/*v*) and stored at −20 °C.

### 3.7. Linearity of SRM Analysis in GD1a Analysis

GD1a (d18:1/18:0) standards were diluted to 0.1, 0.3, 1, 3, 10, 30, 100, 300, and 1000 ng/mL with water/ethanol (1:3, *v*/*v*), and 50 µL of aliquot was injected into the LC–MS/MS system. The peak areas in the SRM analysis were plotted to confirm the correlation between concentration and peak area.

### 3.8. Investigation of Solvent for Redissolution of Targeted GSL Analysis Using GD1a (d18:1/18:0) Standard Solution

GD1a (d18:1/18:0) (100 µL) standard solution was evaporated to dry and redissolved in 100 µL of 6 solvents: 20% methanol, 50% methanol, 75% ethanol, methanol, ethanol, and isopropanol, and 50 µL of each aliquot were used for LC–MS/MS analysis. The peak areas of GD1a (d18:1/18:0) were compared without drying the standard solution.

### 3.9. Investigation of Pretreatment Using GD1a (d18:1/18:0) Standard Solution

Pretreatment for GSL analysis was performed using a combination of liquid–liquid extraction (LLE) and PP using a GD1a (d18:1/18:0) standard solution. First, LLE was performed using the modified Bligh and Dyer method [52,53,57]. To 100 μL of 100 ng/mL GD1a (d18:1/18:0) standard solution in 75% ethanol, 1 mL of water, 1 mL of methanol, and 2 mL of chloroform were added and mixed. The mixture was centrifuged at 3000× *g* for 5 min at 4 °C and separated into two layers. The upper layer was dried in a centrifugal evaporator at 40 °C, and the lower layer was under N_2_ air flow gas dry up at 40 °C. Each aliquot was redissolved in 100 μL of 75% ethanol and analyzed using LC–MS/MS. The peak area of GD1a (d18:1/18:0) was integrated and compared with that without LLE. For PP, 4 mL of acetonitrile was added to 2 mL of the upper layer and mixed thoroughly. The mixture was centrifuged, and the supernatant was dried in a centrifuge evaporator at 40 °C. The residue was redissolved in 100 μL of 75% ethanol, and a 50 μL aliquot was injected into the LC–MS/MS system.

### 3.10. Investigation of Matrix Effect Using GD1a (d18:1/18:0) Standard Solution and Cell Medium

The GD1a (d18:1/18:0) standard solution and ACHN cell medium were used to study the matrix effect. A 10 ng/mL standard solution of GD1a (d18:1/18:0) and cell culture medium were used to evaluate the matrix effect. The matrix effect was evaluated as an MF [54,55,58]. Pretreatment with LLE and PP was performed sequentially for the three samples. Briefly, to 100 μL of 10 ng/mL GD1a (d18:1/18:0) standard in 75% ethanol, 1 mL water, 1 mL methanol, and 2 mL chloroform were added and mixed. The mixture was centrifuged at 3000× *g* for 5 min at 4 °C and separated into two layers. The upper layer was transferred to another tube and mixed with acetonitrile (4 mL). The mixture was centrifuged at 3000× *g* for 5 min at 4 °C, and the supernatant was dried in a centrifuge evaporator at 40 °C. The residue was redissolved in 100 μL of 75% ethanol, and a 50 μL aliquot was injected into the LC–MS/MS system. The sample sets were as follows: (i) GD1a (d18:1/18:0) 10 ng/mL standard solution, (ii) ACHN cell medium, and (iii) GD1a (d18:1/18:0) 10 ng/mL standard-solution-spiked ACHN cell medium. The MF of GD1a (d18:1/18:0) was calculated using the following formula:$$Matrix\;factor\left(\%\right)=\frac{The\;peak\;area\;of\left(iii\right)-the\;peak\;area\;of\left(i\right)}{The\;peak\;area\;of\;(ii)}\times 100$$

### 3.11. Cell Culturing and Sample Preparation

Cells were cultured as previously described [59]. ACHN (CRL-1611, ATCC, Manassas, VA, USA) cells and human kidney proximal tubule cell line (HK-2, CRL-2019, ATCC) were grown in Dulbecco’s modified Eagle’s medium (Thermo Fisher Scientific, Waltham, MA, USA) supplemented with 10% FBS and 1% penicillin–streptomycin mixed solution (Nacalai Tesque, Kyoto, Japan). Each cell line was seeded at 1 × 1,000,000 cells/dish for GSL analysis. After seeding for 2 days and collecting the medium, 1 mL of the cells were washed twice with 2 mL ice-cold PBS and subsequently collected from the Petri dish using a scraper. Finally, the suspended cells were counted and analyzed using LC–MS/MS. The cell extract (1 mL) in PBS or medium was added to the pretreatment method described above. The supernatant was dried, redissolved in 100 µL of 75% ethanol, and 50 µL was used for GSL analysis.

### 3.12. Urine Samples

Urine samples were collected from both RCC patients and healthy controls. All RCC patients were histopathologically diagnosed at the Department of Urology, Tohoku University Hospital. Urine samples were stored at −80 °C until further analysis. Informed consent was obtained from all urine donors according to the ethical guidelines of the Tohoku University Graduate School of Medicine in 2018 (approval number 2017-1-975).

### 3.13. Urine Pretreatment

To 100 μL of urine, 1 mL of methanol and 2 mL of chloroform were added, mixed, and centrifuged at 3000× *g* for 10 min at 4 °C. The supernatant was collected, mixed with 4 mL of acetonitrile, and centrifuged again under the same conditions. The supernatant was dried, redissolved in 100 µL of 75% ethanol, and 50 µL was used for GSL analysis.

## 4. Conclusions

In this study, we applied a semi-quantitative, wide-targeted analysis of GSLs in RCC cells and the urine of RCC patients. We successfully developed a widely targeted GSL analysis method using negative LC–MS/MS SRM analysis. We applied this method to cell samples and found that DSGb5 (d18:1/16:0) expression in the ACHN medium was approximately five times higher than that in the HK-2 cell medium. Finally, we preliminarily analyzed urine samples and found that DSGb5 (d18:1/16:0) in the urine of RCC patients was significantly higher than that in the urine of healthy controls. In the ROC analysis, DSGb5 (d18:1/16:0) was estimated to have a good screening performance for RCC. Therefore, accurate quantification of DSGb5 (d18:1/16:0) is required for precise RCC screening. In the future, we plan to evaluate the screening performance of GSLs, particularly DSGb5, in RCC detection and classification. In addition, we will discuss the implications of our findings in understanding RCC pathogenesis and developing novel diagnostic and therapeutic strategies.

## Figures and Tables

**Figure 1 ijms-25-04098-f001:**
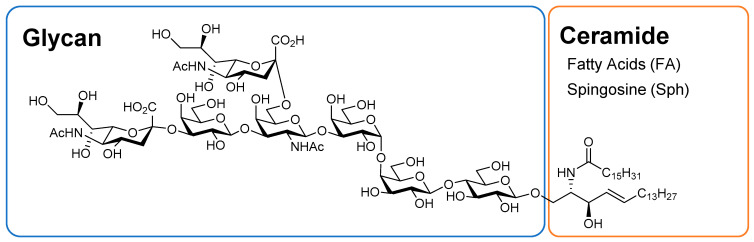
Representative glycosphingolipid structure. This structure shows disialosyl globopentaosylceramide (DSGb5).

**Figure 2 ijms-25-04098-f002:**
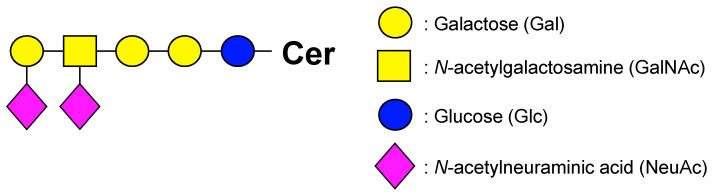
Symbol Nomenclature for Glycans (SNFG): Cer, ceramide. This symbol shows the DSGb5.

**Figure 3 ijms-25-04098-f003:**
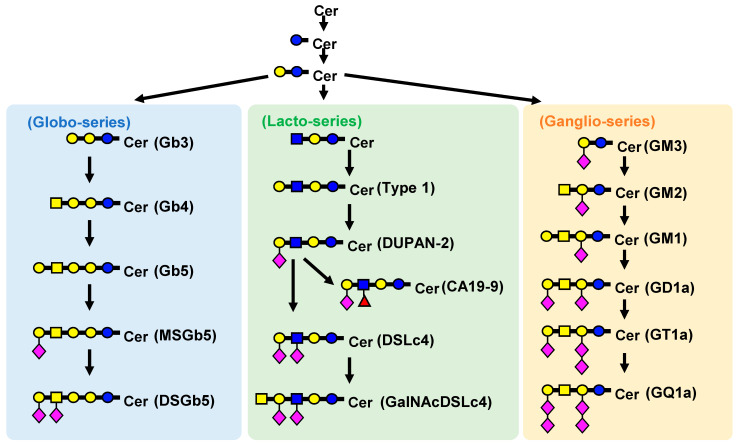
Biosynthetic pathway of GSLs: MSGb5, monosialosyl globopentaosylceramide; DSGb5, disialosyl globopentaosylceramide; GM3, monosialodihexosylganglioside; GM2, monosialotrihexosylganglioside; GM1, monosialotetrahexosylganglioside; DUPAN-2, Duke pancreatic monoclonal antigen type 2; GD1a, ganglioside disialic acid 1a; DSLc4, α(2,3)/α(2,6) disialyl lactotetraosylceramide; GalNAcDSLc4, *N*-acetylgalactosaminyl disialyl lactotetraosyl ceramide; CA19-9, carbohydrate antigen 19-9. The symbols are notated in the same way as in Figure 2.

**Figure 4 ijms-25-04098-f004:**
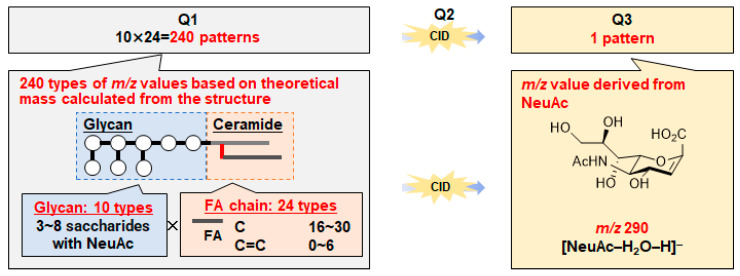
In silico predicted SRM conditions for wide-targeted GSL analysis. For SRM analysis, 240 transitions were set for GSLs consisting of 10 glycans and 24 FA chains. The *m*/*z* of Q1 was calculated from the combination of ceramide and glycans, and Q3 was set at *m*/*z* 290, derived from sialic acid: CID, collision-induced dissociation; FA, fatty acid; SRM, selected reaction monitoring.

**Figure 5 ijms-25-04098-f005:**
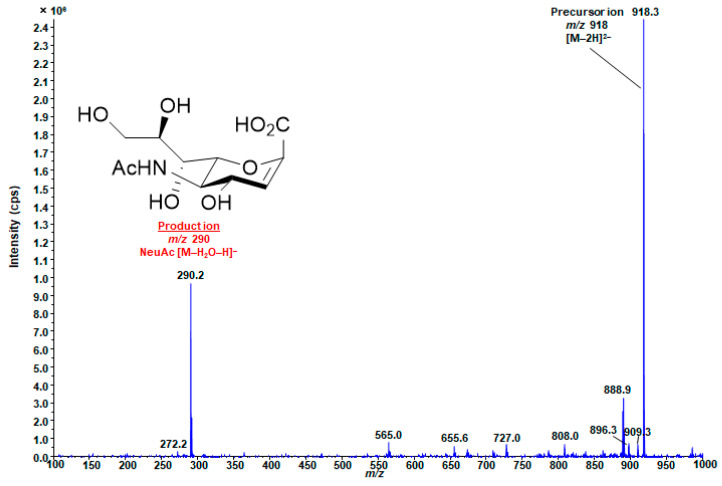
Representative product ion spectrum of GD1a (d18:1/18:0). *m*/*z* 918 is [M-2H]^2−^ of GD1a (d18:1/18:0). *m*/*z* 290, which is derived from NeuAc moiety, was detected as the most intense product ion: GD1a, ganglioside disialic acid 1a.

**Figure 6 ijms-25-04098-f006:**
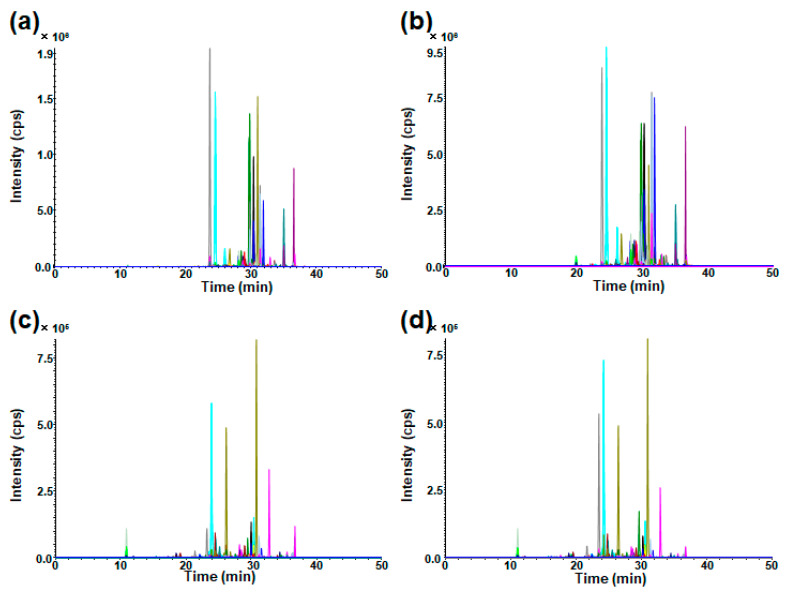
Typical SRM chromatograms for wide-targeted GSL analysis: (**a**) HK-2 cells, (**b**) ACHN cells, (**c**) HK-2 medium, and (**d**) ACHN medium. Each colored SRM chromatogram corresponds to a theoretical GSL structure: SRM, selected reaction monitoring.

**Figure 7 ijms-25-04098-f007:**
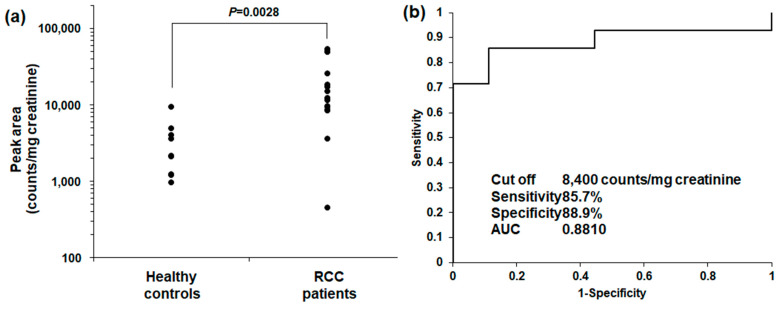
Screening performance of urinary DSGb5 levels (d18:1/16:0): (**a**) DSGb5 (d18:1/16:0) peak intensities in the urine of healthy controls and RCC patients; (**b**) ROC analysis and the performance status of urinary DSGb5 levels (d18:1/16:0). The urinary DSGb5 (d18:1/16:0) peak intensity was significantly higher in RCC patients than that in healthy controls. The screening performance of DSGb5 (d18:1/16:0) was very high.

**Table 1 ijms-25-04098-t001:** Intense GSL peaks with more than 2,000,000 counts in the medium.

No	Component	Q1 (*m*/*z*)	HK-2 (Counts)	ACHN (Counts)	Ratio (%)
1	DSGb5 (d18:1/16:0)	984.5	997,593	4,577,916	459
2	GD1a (d18:1/16:0)	903.5	6,356,833	7,462,803	117
3	GD1a (d18:1/18:0)	917.5	4,503,695	4,343,382	96.4
4	GM3 (d18:1/16:0)	1151.7	6,482,229	5,807,321	89.6

The ratio was calculated as the intensity of each GSL in the ACHN cell medium to that in the HK-2 medium.

## Data Availability

Data are contained in the article and Appendix A.

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
