# Peer review of "Wide-Targeted Semi-Quantitative Analysis of Acidic Glycosphingolipids in Cell Lines and Urine to Develop Potential Screening Biomarkers for Renal Cell Carcinoma"

_ijms, 2024, doi:10.3390/ijms25074098_

Round 1

Reviewer 1 Report

Comments and Suggestions for Authors

Dear editors:  

 It is a great honor and pleasure for me to be invited as the reviewer for this important work entitled “Wide-Targeted Semi-Quantitative Analysis of Acidic Glycosphingo-lipids in Cell Lines and the Urine to Develop Potential Diagnostic Biomarkers for Renal Cell Carcinoma”.  Masamitsu Maekawa and co-authors investigated the development of novel diagnostic biomarkers for Renal Cell Carcinoma. This study topic is novel and interesting, attributing to their team’s long-term efforts and contributions in this scientific field. I have a number of comments concerning this study:

1.     Table 1: Ratio (%) showed the results of 459 and 117 that seem unreasonable.

2.     Line 246: The optimized parameters were shown in Table ??

3.     The table footnote and figure legend should be provided appropriately, e.g., abbreviations in Table 1.

4.     The author should provide more bio-demographic characteristics of patients for comparison in a table, including the flowchart of enrollment process.

5.     From the perspective of a clinician, an AUC of 0.89 for diagnosis is suboptimal that should not be used for cancer diagnosis. Cancer diagnosis is very serious and followed by tumor/organ resection and various intensive therapies. Legal problems often occur due to inconsistent results between the tissue pathology report and the novel diagnostic tool. Thus, all the terms of diagnosis in the study should be replaced by screen.

The research is interesting that could be published after appropriate revision.

Comments on the Quality of English Language

Extensive editing of English language is required.

Author Response

Author's Reply to the Review Report (Reviewer 1)
Dear editors:
It is a great honor and pleasure for me to be invited as the reviewer for this important work entitled “Wide-Targeted Semi-Quantitative Analysis of Acidic Glycosphingo-lipids in Cell Lines and the Urine to Develop Potential Diagnostic Biomarkers for Renal Cell Carcinoma”.  Masamitsu Maekawa and co-authors investigated the development of novel diagnostic biomarkers for Renal Cell Carcinoma. This study topic is novel and interesting, attributing to their team’s long-term efforts and contributions in this scientific field. I have a number of comments concerning this study:
Answer:
Thank you for your kind review. We reconsidered the manuscript and revised our manuscript. We would like to ask you to confirm them.

1.     Table 1: Ratio (%) showed the results of 459 and 117 that seem unreasonable.
Answer:
Thank you for your kind review. This result might mean that the DSGb5 is a GSL type of increasing in RCC pathology than other GSLs. Therefore, we added the description about the significance of increasing to line 194. And, the footnote of Table 1 was added as well.

2.     Line 246: The optimized parameters were shown in Table ??
Answer:
Thank you for your kind review. The optimized parameters were shown in Table S2 and so, the description was added in the text.

3.     The table footnote and figure legend should be provided appropriately, e.g., abbreviations in Table 1.
Answer:
Thank you for your kind review. As you pointed out, we added the table footnote and figure legend.

4.     The author should provide more bio-demographic characteristics of patients for comparison in a table, including the flowchart of enrollment process.
Answer:
Thank you for your kind review. We added the individual data to Table S7 and demographic data to Table S8. In this study, we were not able to collect the age-matched urine samples and so, we tried to investigate the another cohort for validating this study’s result. And, we added the description about enrollment process to Section “3.12. Urine samples”. 

5.     From the perspective of a clinician, an AUC of 0.89 for diagnosis is suboptimal that should not be used for cancer diagnosis. Cancer diagnosis is very serious and followed by tumor/organ resection and various intensive therapies. Legal problems often occur due to inconsistent results between the tissue pathology report and the novel diagnostic tool. Thus, all the terms of diagnosis in the study should be replaced by screen.
Answer:
Thank you for your kind review. As you pointed out, we replaced the words “diagnosis” to “screen”. 

The research is interesting that could be published after appropriate revision.
Answer:
Thank you for your kind review.

Reviewer 2 Report

Comments and Suggestions for Authors

Manuscript ijms-2915190

Comments

The authors of this paper established a protocol for targeted glycosphingolipid (GSL) analysis by LC-MS, specifically for use with renal cell carcinoma, in order to identify possible GSL biomarkers. They compared HK-2 cell line (as control) with the ACHN cell line. They identified the GSL DSGb5 as significantly increased in ACHN cell medium. Most importantly, the authors could confirm elevated DSGb5 levels in urine of RCC patients.

However, presence of DSGb5 in ACHN cells and its likely correlation with higher malignant potential has been shown in earlier publications (by some of the authors of the present manuscript). Thus, the new information provided is the detectable increase of DSGb5 in urine samples from patients. This is a relevant new information and could have diagnostic potential. The authors, however, should discuss more the earlier publications on this topic and emphasize more what their current study adds to the previous knowledge.

Many data/informations are presented in supplementary informations. I would suggest to move some of them to the main manuscript. At least the structure of DSGbB5 and/or the ganglioside biosynthesis (Fig. S1, S3) should be shown in the main manuscript (not only in supplement)

Supplementary Figures (Power Point file):

Fig. S1 and S2 gives the structure of DSGb5; however the Figure legends stated „Structure of GSLs“, the authors should write „Structure of ganaglioside DSGb5“ instead.

Because there are no data on on patients and control persons, it is not possible to judge whether there are differences in age and sex between the groups; could the authors provide such informations?

Abstract: Please write out all terms in the abstract at first appearance, and if necessary define abbreviations: RCC, ACHN, ROC;

line 184: last sentence is incomplete;

line 307: HK-2 cells have the number CRL-2190 NOT CRL-20190

I suggest to place the Conclusions Section after the Results/Discussion, before the Methods Section.

Author Response

Author's Reply to the Review Report (Reviewer 2)
Comments
The authors of this paper established a protocol for targeted glycosphingolipid (GSL) analysis by LC-MS, specifically for use with renal cell carcinoma, in order to identify possible GSL biomarkers. They compared HK-2 cell line (as control) with the ACHN cell line. They identified the GSL DSGb5 as significantly increased in ACHN cell medium. Most importantly, the authors could confirm elevated DSGb5 levels in urine of RCC patients.
However, presence of DSGb5 in ACHN cells and its likely correlation with higher malignant potential has been shown in earlier publications (by some of the authors of the present manuscript). Thus, the new information provided is the detectable increase of DSGb5 in urine samples from patients. This is a relevant new information and could have diagnostic potential. The authors, however, should discuss more the earlier publications on this topic and emphasize more what their current study adds to the previous knowledge.
Answer:
Thank you for your kind review. As you pointed out, we added the descriptions to Section 2.2 and 2.3.

Many data/informations are presented in supplementary informations. I would suggest to move some of them to the main manuscript. At least the structure of DSGbB5 and/or the ganglioside biosynthesis (Fig. S1, S3) should be shown in the main manuscript (not only in supplement)
Answer:
Thank you for your kind review. We moved 3 supplementary Figure (Fig. S1-3) to main figure 1-3.

Supplementary Figures (Power Point file):

Fig. S1 and S2 gives the structure of DSGb5; however the Figure legends stated „Structure of GSLs“, the authors should write „Structure of ganaglioside DSGb5“ instead.
Answer:
Thank you for your kind review. We revised the Fig. S1 and S2 (moved to Fig. 1 and 2) legend.

Because there are no data on on patients and control persons, it is not possible to judge whether there are differences in age and sex between the groups; could the authors provide such informations?
Answer:
Thank you for your kind review. We added the patient individual data to Table S7 and summarized demographic data to Table S8. 

Abstract: Please write out all terms in the abstract at first appearance, and if necessary define abbreviations: RCC, ACHN, ROC;
Answer:
Thank you for your kind review. We added the spelling of RCC and ROC. But we could not add the spelling of ACHN, because ACHN is cell line name.

line 184: last sentence is incomplete;
Answer:
Thank you for your kind review. We added the description. 

line 307: HK-2 cells have the number CRL-2190 NOT CRL-20190
Answer:
Thank you for your kind review.

I suggest to place the Conclusions Section after the Results/Discussion, before the Methods Section.
Answer:
Thank you for your kind review. This journal has to place Conclusion Section after Methods Section.

Reviewer 3 Report

Comments and Suggestions for Authors

In the manuscript entitled "Wide-Targeted Semi-Quantitative Analysis of Acidic Glycosphingolipids in Cell Lines and the Urine to Develop Potential Diagnostic Biomarkers for Renal Cell Carcinoma", the authors provided insight into new possibilities for timely, non-invasive diagnosis of renal cell carcinoma. For this purpose, they used modern diagnostic methods to reveal the connection between the disease and the presence of certain biomarkers in the examined samples. The scientific approach to this problem is appropriate and the chosen methodology is adequate. Because of all this, this research has the potential to indicate new diagnostic approaches that could possibly have a positive effect on the outcome of the aforementioned disease.

In the following, I would like to refer to several observations, in the order in which they are encountered in the manuscript:

Line 62: de novo (italic)

The two sentences describing the method (lines 112, 113, 114) are redundant. The method is well known so it is not necessary to describe its general features.

Line 132: in silico (italic)

The sentence started in line 163 seems to be redundant since the data in it was just repeated from the previous sentence. Additionally, it seems to me that the first sentence describing the results (Line 161-163) should be reworded to make it more understandable.

It would be good in the introduction to explain the nature of the ACHN cell line, and thus its connection with the experimental design. Its origin was mentioned in Line 183, but until then it was mentioned several times as an unexplained abbreviation, so the reader who is not familiar with the origin of that cell line has difficulty in understanding the experimental design.

The entire paragraph from lines 178 to 184 should be reworded to make it clearer. Namely, the data from the third sentence is already contained in the first one (which also needs to be edited linguistically), and the last sentence of the paragraph is incomplete (The typical chromatograms were shown in ?????).

Line 234: Two adjacent sentences start with the same words, so it would be good to change them so that the text is more linguistically connected.

Line 241: “In”, lower case, although I think that maybe that whole expression (“In this time”) is not necessary

Line 244: collision gas (CAD), collision gas (CAD) - repetition????

Line 246: The first question is in which table are the mentioned results shown? Another question is why is it mentioned in this chapter (unless it is supplementary material)?

Line 251, 260, 325, 326: ºC is not a symbol for Celsius degrees

Line 268 and 269: It looks like the first two sentences of the paragraph should be merged because the data is repeated.

Line 278: And the mixture was… It is unusual that the sentence starts with AND.

Line 311: "medium 1 mL, the cells were..." should be as follows: "medium, 1 mL the cells were..."

Paragraph 3.12. Urine samples: In the description of the samples, you state that patients were classified according to tumor stage, grade, and histology. However, when looking at the results of the analysis (Table 4), it can be seen that the samples were divided into only two groups, healthy controls and RCC patients. For the purposes of achieving the goal of your experimental design, it is not even necessary to analyze different groups of patients separately (although this would be an informative result regarding the correlation of tumor stage and the studied biomarker), but if you have already mentioned that the patients are categorized, then it is expected that this can be seen in results. Please comment on this observation of mine and think about wording this paragraph differently.

Paragraph 4 Conclusions: This paragraph should be aimed exclusively at emphasizing the scientific and practical significance of your findings. This means that it is not recommended to start it with one lukewarm sentence in which you say that you have tried something. You haven't tried. You recognized the need to improve the diagnosis of a dangerous disease, and then you decided to develop a method for this purpose that involves the use of an appropriate device and type of analysis. And you got promising results that say your idea makes sense. In short, the first sentence is the weak point of this paragraph, so I suggest you think about reformulating the conclusions.

Author Response

Author's Reply to the Review Report (Reviewer 3)

In the manuscript entitled "Wide-Targeted Semi-Quantitative Analysis of Acidic Glycosphingolipids in Cell Lines and the Urine to Develop Potential Diagnostic Biomarkers for Renal Cell Carcinoma", the authors provided insight into new possibilities for timely, non-invasive diagnosis of renal cell carcinoma. For this purpose, they used modern diagnostic methods to reveal the connection between the disease and the presence of certain biomarkers in the examined samples. The scientific approach to this problem is appropriate and the chosen methodology is adequate. Because of all this, this research has the potential to indicate new diagnostic approaches that could possibly have a positive effect on the outcome of the aforementioned disease.

In the following, I would like to refer to several observations, in the order in which they are encountered in the manuscript:

Answer:

Thank you for your kind review. We revised the manuscript by according to your valuable comments. We would like to ask you confirm it.

Line 62: de novo (italic)

Answer:

Thank you for your kind review. We revised the description.

The two sentences describing the method (lines 112, 113, 114) are redundant. The method is well known so it is not necessary to describe its general features.

Answer:

Thank you for your kind review. We integrated two sentences to one sentence.

Line 132: in silico (italic)

Answer:

Thank you for your kind review. We revised the description.

The sentence started in line 163 seems to be redundant since the data in it was just repeated from the previous sentence. Additionally, it seems to me that the first sentence describing the results (Line 161-163) should be reworded to make it more understandable.

Answer:

Thank you for your kind review. We revised the description for understanding easier.

It would be good in the introduction to explain the nature of the ACHN cell line, and thus its connection with the experimental design. Its origin was mentioned in Line 183, but until then it was mentioned several times as an unexplained abbreviation, so the reader who is not familiar with the origin of that cell line has difficulty in understanding the experimental design.

Answer:

Thank you for your kind review. We added the description about purpose of using ACHN cell and the strategy of finding GSL species for potential RCC screening biomarkers.

The entire paragraph from lines 178 to 184 should be reworded to make it clearer. Namely, the data from the third sentence is already contained in the first one (which also needs to be edited linguistically), and the last sentence of the paragraph is incomplete (The typical chromatograms were shown in ?????).

Answer:

Thank you for your kind review. We deleted the redundant description and added the description about incomplete sentence.

Line 234: Two adjacent sentences start with the same words, so it would be good to change them so that the text is more linguistically connected.

Answer:

Thank you for your kind review. We revised the description to connect the sentences.

Line 241: “In”, lower case, although I think that maybe that whole expression (“In this time”) is not necessary

Answer:

Thank you for your kind review. We deleted the description.

Line 244: collision gas (CAD), collision gas (CAD) - repetition????

Answer:

Thank you for your kind review. We deleted the one CAD description.

Line 246: The first question is in which table are the mentioned results shown? Another question is why is it mentioned in this chapter (unless it is supplementary material)?

Answer:

Thank you for your kind review. As you pointed out, the result should be shown in Result Section. And so, the sentence was deleted.

Line 251, 260, 325, 326: ºC is not a symbol for Celsius degrees

Answer:

Thank you for your kind review. We replaced degree sign.

Line 268 and 269: It looks like the first two sentences of the paragraph should be merged because the data is repeated.

Answer:

Thank you for your kind review. We deleted the redundant description.

Line 278: And the mixture was… It is unusual that the sentence starts with AND.

Answer:

Thank you for your kind review. We deleted “AND”.

Line 311: "medium 1 mL, the cells were..." should be as follows: "medium, 1 mL the cells were..."

Answer:

Thank you for your kind review. We revised the description by according to your comments.

Paragraph 3.12. Urine samples: In the description of the samples, you state that patients were classified according to tumor stage, grade, and histology. However, when looking at the results of the analysis (Table 4), it can be seen that the samples were divided into only two groups, healthy controls and RCC patients. For the purposes of achieving the goal of your experimental design, it is not even necessary to analyze different groups of patients separately (although this would be an informative result regarding the correlation of tumor stage and the studied biomarker), but if you have already mentioned that the patients are categorized, then it is expected that this can be seen in results. Please comment on this observation of mine and think about wording this paragraph differently.

Answer:

Thank you for your kind review. As you pointed out, we reconsidered the purpose of this study. As you show, we don’t aim to investigate the relationship various RCC classification and GSL abundance. Therefore, we revised the description simply.

Paragraph 4 Conclusions: This paragraph should be aimed exclusively at emphasizing the scientific and practical significance of your findings. This means that it is not recommended to start it with one lukewarm sentence in which you say that you have tried something. You haven't tried. You recognized the need to improve the diagnosis of a dangerous disease, and then you decided to develop a method for this purpose that involves the use of an appropriate device and type of analysis. And you got promising results that say your idea makes sense. In short, the first sentence is the weak point of this paragraph, so I suggest you think about reformulating the conclusions.

Answer:

Thank you for your kind review and valuable comments. As you pointed out, we reconsidered the significance of this study. As you show, we revised the first sentence of this Section.

Round 2

Reviewer 1 Report

Comments and Suggestions for Authors

The authors work hard to respond to the reviewers' comments.

Reviewer 2 Report

Comments and Suggestions for Authors The authors have satisfactorily addressed all critical points.

Reviewer 3 Report

Comments and Suggestions for Authors

Dear authors, thank you for your cooperation during the revision of your manuscript.

Best regards.